# Advanced Simulation System for Orbitozygomatic Fracture Reconstruction: Multicenter Validation of a Novel Training and Objective Assessment Platform

**DOI:** 10.3390/cmtr18030034

**Published:** 2025-08-14

**Authors:** Enrique Vargas, Rodrigo Díaz, Juan Pablo Vargas, Andrés Campolo, Rodrigo Villanueva, Carlos Cortéz, Salvador Valladares-Pérez

**Affiliations:** 1School of Dentistry, Pontificia Universidad Catolica de Chile, Santiago 7820436, Chile; rdiag@uc.cl (R.D.); jpvargas@uc.cl (J.P.V.); afcampolo@uc.cl (A.C.); rrvillanueva@uc.cl (R.V.); carlos.cortez@uc.cl (C.C.); salvadorvalladares@gmail.com (S.V.-P.); 2Metropolitan Clinical Hospital El Carmen, Maipu, Santiago 9251521, Chile

**Keywords:** surgical education, technical skills, synthetic surgical model, performance feedback, patient safety, OSATS

## Abstract

Orbitozygomatic fractures represent a complex surgical challenge. Given the urgent need for validated educational tools that surpass traditional learning models, this multicenter study developed and validated a novel synthetic advanced simulation model for the reconstruction of these fractures. The model integrates platinum-cured silicones and 3D-printed bony structures with prefabricated fractures, accurately replicating the anatomy and tactile properties of soft and hard tissues, including simulated herniation of orbital contents. To our knowledge, it is the only available synthetic model combining both tissue types for this training. Ten participants (faculty and residents) completed simulated procedures. Technical performance was assessed using a hand motion tracking system, the global OSATS (Objective Structured Assessment of Technical Skills) scale, and a task-specific error measurement (Specific Fault Measurement, SFM) scale. Statistically significant differences (*p* = 0.021) were observed in operative time and error count between novices and experts, confirming the model’s construct validity. Faculty completed the surgery in significantly less time (mean 18.16 min vs. 37.01 min for residents) and made fewer errors (mean 12.25 vs. 53.25). Face and content validity were strongly supported by participant surveys, with 100% stating they would use the simulator to practice before real surgery. A strong inverse correlation (r = –0.786, *p* = 0.021) between OSATS and SFM scores demonstrated concurrent validity. This model enables ethical, repeatable, and cost-effective training, supporting its implementation into surgical curricula to enhance competence and provide objective skill assessment in orbitozygomatic trauma surgery.

## 1. Introduction

According to facial architecture, a direct impact on the eye and/or the orbital rim with sufficient energy to cause fractures typically results in fractures occurring in areas of least resistance [1], primarily the orbital floor and medial wall. However, due to the dynamics of traumatic forces, it is common for the lateral wall to be involved, resulting from a zygomatic displacement that compromises the fronto-zygomatic and spheno-zygomatic sutures.

High-energy trauma (such as motor vehicle accidents) has increased the incidence of this type of fracture [2]. Post-traumatic enophthalmos remains one of the most challenging complications to treat [3], and of particular importance are patients with an orbital floor fracture and an associated inferior rectus entrapment, as this condition constitutes a surgical emergency.

Complex orbital fractures present significant surgical challenges due to the presence of multiple anatomical structures within the orbital cavity and the limited visibility during surgery [3]. As such, the operative management of facial trauma requires a high level of clinical expertise and advanced decision-making skills [4].

Reconstruction of orbitozygomatic fractures demands not only surgical proficiency but also specialized training in minimally invasive techniques [5,6], which are essential for restoring function while minimizing aesthetic sequelae in the facial region [7].

In this context, surgical simulation has been demonstrated to be an efficient and effective tool for acquiring surgical skills. It offers a safe and realistic environment where trainees can practice surgical techniques, thereby reducing patient risk and improving the confidence and performance of surgical residents [8,9,10]. Considering that simulation-based teaching tools have proven effective in various educational settings, their incorporation into surgical training programs has become increasingly essential [11,12].

Previous educational models for orbitozygomatic anatomy [13], soft tissue management [14,15,16,17,18], and fracture reconstruction [19,20] have been developed for training purposes. However, to date, soft tissue models lack the capability to simulate deep orbital structures, and 3D-printed orbital models remain insufficient, as they have not been able to replicate the experience of reducing entrapped orbital soft tissues [19].

To our knowledge, no synthetic simulation models combining both soft and hard tissues are currently available for training residents in orbitozygomatic fracture reconstruction techniques [8]. Given that proper handling of soft tissues is crucial to achieving optimal aesthetic outcomes, specialized training is required. Simulation models provide an opportunity for deliberate practice [9], an objective and structured form of training that includes real-time feedback and unbiased evaluation [4].

Thus, we present the development and validation of a novel high-fidelity simulation model and training program through a simulation-based research design in which simulation serves as the subject of investigation rather than merely as a research method. The objectives of this study are to assess face and construct validity, validate a specific rating scale for each task, and describe the development of a high-fidelity synthetic orbitozygomatic reconstruction training model for maxillofacial trainees.

The primary objective of this study is to develop a synthetic simulation model and associated training program for orbitozygomatic fracture reconstruction, with a focus on evaluating its face, content, and construct validity. To achieve this, this study has several specific aims: first, to test the simulation model with a sample of faculty and residents (*n* = 10) and to determine whether there are statistically significant differences in performance between the two groups; second, to evaluate face and content validity through a structured questionnaire administered to participants; and third, to assess construct validity by measuring surgeons’ hand motion parameters using the ICSAD (Imperial College Surgical Assessment Device), evaluating overall surgical performance with the 5-point Likert-based OSATS (Objective Structured Assessment of Technical Skills) scale, and quantifying technical errors in key surgical domains through expert evaluation and a complementary error-specific measurement system.

## 2. Materials and Methods

This study was written in accordance with the reporting guidelines for simulation-based research, extensions of the CONSORT and STROBE statements [21].

### 2.1. Simulation Model Construction

The development of the simulation model began by deconstructing orbitozygomatic fracture reconstruction surgery into essential procedural tasks that trainees must master within the training program. These tasks were organized into a standardized surgical protocol, designed to be applied consistently across all evaluated procedures. Two expert maxillofacial surgeons from the Faculty of Medicine at Pontificia Universidad Católica de Chile identified the critical stages of the reconstruction and established a detailed protocol encompassing all steps necessary to complete the surgery, from incision to final closure.

Based on this protocol—and supported by 3D image processing techniques, as well as direct observation of anatomical dissections and live surgical procedures (Figure 1)—a synthetic model was developed to incorporate expert-driven anatomical and procedural recommendations (Figure 2 and Figure 3).

To replicate the anatomical characteristics of the human orbitozygomatic region, 3D printing technology was used to fabricate negative molds that accurately reproduced the morphology of both hard and soft midfacial tissues. The digital models were designed and processed using the following 3D modeling and slicing software: Blender 4.2 LTS, Autodesk Fusion 360, Autodesk Meshmixer 3.5.0, Ultimaker Cura 5.7.2, and Bambu Studio v2.1.1. Physical fabrication was performed using Artillery Sidewinder X1 (Artillery, Shenzhen, China), Creality CR-10 Pro (Creality, Shenzhen, China), and Bambu Lab A1 (Bambu Lab, Shenzhen, China) 3D printers, selected based on convenience and the specific technical requirements of each component.

The simulation of hard tissues was achieved using polyurethane-12, while the soft tissues were recreated through incremental layering of platinum-cured silicones, ensuring anatomical and tactile fidelity. The resulting model replicated with high precision the anatomical structures of the midface, particularly the periorbital region and its components. The simulation included representations of the skin, subcutaneous tissue, fat, muscle, periosteum, sclera, conjunctiva, and the infraorbital neurovascular bundle. Each layer was engineered to exhibit the appropriate consistency, color, elasticity, and mechanical resistance of its real-life counterpart. The soft tissues were fabricated using platinum-catalyzed silicone, as follows. For the eyeball, Ecoflex 00:10 silicone (Smooth-On; Macungie, PA, USA) was used with a 20:1 ratio of white to yellow pigment for the sclera. Periorbital fat was simulated using Ecoflex Gel (Smooth-On; Macungie, PA, USA) with 20% Slacker and a 10:1 ratio of yellow to white pigment. Muscle tissue was created using Ecoflex 00:50 (Smooth-On; Macungie, PA, USA) silicone pigmented in blood red. Subcutaneous tissue was produced using Ecoflex Gel with a 5:1 ratio of blood-red to skin-tone pigment. Finally, the skin layer was fabricated using 5% Ecoflex 00:50 silicone with skin-tone pigment. Furthermore, prefabricated fractures were incorporated into the frontozygomatic suture and orbital floor to ensure uniformity across models and to simulate the herniation of orbital contents into the maxillary sinus.

This design enables trainees to become familiar with the use of osteosynthesis materials—including orbital floor mesh, plates, screws, and the corresponding surgical instruments—but, more importantly, it allows them to simultaneously integrate proper technique in soft tissue handling while coordinating all these elements in a cohesive manner. This comprehensive approach significantly enhances both the realism and the educational effectiveness of the surgical training experience.

### 2.2. Ethics and Participants

This study was formally reviewed and approved by the Institutional Review Board of the Faculty of Medicine at Pontificia Universidad Católica de Chile, Santiago, Chile (approval ID: 230809001).

Prior to participation, all individuals enrolled in the training program completed an initial questionnaire aimed at collecting background information regarding their surgical experience and prior exposure to simulation-based training. The full questionnaire is presented in Table 1.

Sample size estimation was informed by previous validation studies, indicating a minimum requirement of three participants per group (trainees and non-trainees) [22]. Based on a standard deviation of 3.1 and a mean difference of 13.5 from previous findings, a power analysis indicated that at least three experts and three non-expert participants were required to achieve sufficient statistical power [23].

Participants were classified into two groups according to their clinical experience: trainees and faculty. The trainee group consisted of maxillofacial surgery residents in postgraduate years 2 through 4, each having participated in no more than three orbitozygomatic fracture reconstruction procedures. The faculty group included board-certified maxillofacial surgeons with clinical teaching experience; each having performed at least 30 orbitozygomatic reconstruction procedures. All faculty members were affiliated with one of three leading Chilean institutions recognized for their expertise in maxillofacial trauma: Hospital Clínico Mutual de Seguridad (CChC), Hospital de Carabineros (HOSCAR), and Pontificia Universidad Católica de Chile (PUC).

Exclusion criteria included any interruption of clinical activities for a period exceeding six months within the last two years.

Before participating in the simulation session, all individuals completed a standardized orientation session, filled out a demographic survey, and provided written informed consent.

### 2.3. Simulation Assessment

The assessment of the simulation training program was conducted in five phases:Simulated surgeries were performed, during which performance data was collected for each participant. Procedures were video-recorded and assessed with ICSAD to measure performance metrics and evaluate construct validity through motion analysis.Immediately following the procedures, participants completed structured satisfaction surveys aimed at evaluating the face and content validity of the simulation model.Postoperative CBCT evaluation by a blinded radiologist assessed six key aspects using standardized criteria.In the fourth stage, two expert surgeon evaluators, blinded to participant identity and group allocation, reviewed the performance data collected during the first stage. Each participant was assessed using both a global rating scale (OSATS) and a specific objective, fault-based scale (Specific Fault Measurement, SFM) to evaluate technical performance.In the final phase, statistical comparisons were conducted between groups to determine whether significant differences in performance outcomes could be identified.

#### 2.3.1. First Stage: Simulated Surgeries

Participants were paired in dyads, each composed of one trainee and one faculty member, emulating the traditional surgical training model in which an expert supervises and guides a novice through a clinical procedure [24]. In this format, the expert provided real-time guidance to address any gaps in knowledge not explicitly covered by the standardized surgical protocol.

Both participants worked on identical synthetic simulators and were instructed to follow the same standardized surgical protocol to treat an orbitozygomatic complex fracture, which included a blowout fracture of the orbital floor and a frontozygomatic suture fracture. The reconstruction was performed using a minimally invasive technique, employing real surgical instruments on the simulation model.

Construct validity, defined as the simulation system’s ability to distinguish between novice and expert performance under standardized, controlled conditions, was assessed by evaluating whether the system could identify performance differences between trainees and faculty—and whether trainees could, over time, approach expert-level proficiency.

In this study, all participants were instructed to follow the procedural steps of the standardized surgical protocol without prior rehearsal.

Each simulation was timed and video-recorded from at least two perspectives, including a first-person point of view and a fixed-camera angle, to ensure comprehensive visualization of the procedure.

Meanwhile, motion tracking data was simultaneously collected using the ICSAD (Imperial College Surgical Assessment Device). Sensors placed on the dorsum of participants’ hands (beneath surgical gloves) tracked the position and movement of both hands across three spatial axes.

These metrics were obtained using the ICSAD system, a validated tool for surgical motion analysis that captures spatial and temporal hand performance parameters via inertial sensors [25,26].

Key performance metrics were recorded, including total operative time (in seconds), path length traveled by both hands (in meters), and number of hand movements.

#### 2.3.2. Second Stage: Structured Satisfaction Surveys

##### Face and Content Validity

The survey consisted of 12 structured items designed to evaluate the simulator’s fidelity in replicating real-life surgical procedures. All items were rated using a 5-point Likert scale, with higher scores indicating greater agreement or satisfaction.

This instrument was specifically developed to measure Level 1 (Reaction) of Kirkpatrick’s Four-Level Model of training evaluation [27], providing information about participants’ perceptions of realism, educational value, and overall satisfaction with the simulation-based experience.

The Face Validity Survey (Table 2) evaluated the anatomical accuracy and realism of the simulator, assessing whether it provided a convincing surgical environment.The Content Validity Survey (Table 3) assessed the simulator’s ability to effectively convey the procedural knowledge and technical skills necessary for a successful orbitozygomatic fracture reconstruction.

#### 2.3.3. Third Stage: Cone-Beam Tomography Ancillary Assessment (CBCT)

Postoperative CBCT was captured (Figure 4) and an objective evaluation was conducted by an experienced maxillofacial radiologist, who served as a blinded assessor. The radiologist systematically assessed six key postoperative outcome aspects in the models using standardized criteria (Table 4), indicating whether each criterion was satisfactorily met.

#### 2.3.4. Fourth Stage: Evaluation by Expert Surgeons

After the simulation, blinded expert raters—selected from the original faculty group—evaluated each performance using two complementary tools:The OSATS (Objective Structured Assessment of Technical Skills) scale, a 5-point Likert scale evaluating overall surgical competence.The Specific Fault Measurement (SFM) system records and quantifies each procedural error throughout the entire surgery, within the same five defined domains evaluated globally by the OSATS framework.

The evaluators were blinded to participant identity and group assignment and were previously trained in both the protocol and use of the simulation training system. Evaluations were based on multi-angle video footage of each procedure, including point-of-view recordings, to ensure uninterrupted visualization of the surgical field. Videos contained no identifying information about the participants.

##### OSATS Global Rating Scale (GRS)

Surgical performance was evaluated using the Global Rating Scale (GRS), derived from the widely validated Objective Structured Assessment of Technical Skills (OSATS) framework [28,29]. The scale comprises five key performance domains, each rated on a 5-point Likert scale, where a score of 5 indicates optimal performance in the corresponding domain.

The five domains assessed were

Respect for tissue integrity.Avoidance of unnecessary movements.Absence of hesitant or awkward movements.Maintenance of a steady operative flow.Demonstrated procedural knowledge.

Each evaluation could range from a minimum of 5 points to a maximum of 25 points. Since performance was assessed independently by two expert evaluators, the combined score for each participant could vary between a minimum of 10 points and a maximum of 50 points.

To facilitate interpretation of the results, OSATS scores were further categorized into three performance tiers—low, intermediate, and high—based on a standardized performance conversion scale (Table 5).

##### Specific Fault Measurement (SFM) Scale

To complement the OSATS framework, a novel evaluation tool was developed as part of this training program: The Specific Fault Measurement (SFM) scale. This complementary assessment instrument was designed to quantify specific technical errors in a detailed and objective manner, addressing the same five core surgical principles evaluated by OSATS but through a fault-based, quantitative lens.

Unlike the global judgment required by the OSATS, the SFM provides a time-stamped recording of procedural transgressions throughout the surgical task. Evaluators documented each error as it occurred, noting both the precise time (minutes and seconds) and the specific domain in which it was committed. All errors were recorded in an error log table, enabling a cumulative error count for each of the five domains.

The total number of errors yielded a numerical SFM score, where higher scores indicate poorer performance. This system enhances the granularity and objectivity of the assessment, and it was specifically designed to improve post-procedural debriefing and feedback. By identifying exact moments and areas of deficiency, instructors are better equipped to guide trainees in developing targeted strategies for improvement and preventing the recurrence of the same errors in future sessions.

The SFM scale was thus developed to serve not only as a validation tool but also as a pedagogical aid, promoting reflective learning and facilitating the design of individualized training plans (Table 6).

#### 2.3.5. Fifth Stage: Statistical Comparisons

Statistical comparisons between the trainee and faculty groups were conducted to determine whether the simulator could effectively distinguish between different levels of surgical expertise in search of evidence of construct, face, and content validity.

All data were analyzed using IBM SPSS Statistics, version 27.0.1.0. Descriptive statistics were calculated for all variables. Given the small sample size and the nonparametric nature of the data, construct validity was assessed using the Mann–Whitney U test to identify statistically significant differences in performance between independent groups (trainees vs. faculty). A *p*-value < 0.05 was considered statistically significant.

The relationship between SFM (Specific Fault Measurement) scores and OSATS (Objective Structured Assessment of Technical Skills) scores was assessed using Spearman’s rank correlation coefficient.

To evaluate the internal consistency of the face and content validity questionnaires, Cronbach’s alpha was used as a reliability coefficient.

## 3. Results

A total of 10 participants were recruited for this study, including 6 faculty members (board-certified maxillofacial surgeons) and 4 residents currently enrolled in a maxillofacial surgery training program. Of the total participants, only two reported prior experience with simulation-based training—both from the faculty group, with none of the residents having such experience.

Participant demographic data and clinical experience are summarized (Table 7).

### 3.1. Construct Validity

Assessment of performance quality using the Global Rating Scale (GRS) from the Objective Structured Assessment of Technical Skills (OSATS) revealed significantly higher scores among faculty members compared to trainees.

As shown in Figure 5, 75% of participants in the trainee group fell within the lowest performance tier, while the remaining 25% were categorized in the intermediate stratum. In contrast, 100% of faculty participants scored within the intermediate or high-performance tiers, with an even distribution of 50% in each category.

The faculty group achieved a median GRS score of 19.5 out of 30 (range: 9–23), whereas the trainee group obtained a median score of 11 (range: 9–20). The mean score difference between the groups was 8.5 points.

The Specific Fault Measurement (SFM), developed to assess the simulation model, demonstrated high internal consistency, with a Cronbach’s alpha of 0.8288, indicating strong reliability of the instrument.

Higher SFM scores were observed among trainee participants, whereas all faculty members obtained lower scores, reflecting superior technical performance.

A strong negative correlation was found between SFM and OSATS scores (r = −0.786, *p* = 0.021). Given the significance level (*p* < 0.05), the null hypothesis was rejected (Figure 6).

Further analysis confirmed these findings: the faculty group exhibited a significantly lower number of procedural faults compared to the trainee group. The mean number of faults recorded per procedure was 53.25 (SD = 22.95) for the trainee group, compared to 12.25 (SD = 5.73) for the faculty group (*p* = 0.021). (Figure 7) (Table 8).

These results indicate a statistically significant difference in performance between novice and expert surgeons.

### 3.2. Operative Time Comparison

Faculty members completed the surgical simulation in a significantly shorter time compared to trainees. The mean operative time for the faculty group was 18.16 min (SD = 5.15), whereas for the trainee group, it was 37.01 min (SD = 11.13).

A non-parametric Mann–Whitney U test was conducted to compare operative times between the two groups, revealing a mean difference of 18.85 min. The results confirmed a statistically significant difference, with a *p*-value of 0.021 (*p* ≤ 0.05), allowing for the rejection of the null hypothesis (Figure 8) (Table 9).

These results indicate a statistically significant difference in performance between novice and expert surgeons.

### 3.3. Hand Movement and Path Length Analysis

Data on hand movements and path length, collected via the Imperial College Surgical Assessment Device (ICSAD), were available for 8 out of 10 participants. Two faculty members were excluded from this analysis due to technical difficulties: in one case, the sensor detached from the participant’s hand, and in the other, a voltage drop caused the ICSAD device to stop recording. As a result, statistical comparisons were conducted on data from four trainees and four faculty members.

Faculty members showed greater economy of motion throughout the procedure.

The mean number of hand movements (movements currently displayed) was 3914.75 (SD = 2341.24) in the faculty group, compared to 5850 (SD = 4074.82) in the trainee group (Figure 9).

Likewise, the total path length traveled by both hands was shorter among faculty, with a mean of 413.41 m (SD = 208.1), versus 630.66 m (SD = 372.64) for trainees (Figure 10).

Despite the faculty group showing greater economy of movements with 33.08% fewer movements currently displayed and 34.44% less total path length, a (*p* ≤ 0.05) was not obtained in these categories.

### 3.4. Content and Face Validity

Table 10 and Table 11 present the results of the Likert-scale survey used to evaluate both content and face validity, respectively, detailing participants’ perceptions regarding the realism and educational relevance of the simulation model.

#### 3.4.1. Content Validity

The internal consistency of the content validity questionnaire was evaluated using Cronbach’s alpha, which yielded a value of 0.879, indicating good reliability of the instrument.

As shown in Table 10,

In terms of surgical approach training, 100% of respondents agreed that the simulator could be effectively used to teach access to orbitozygomatic fractures (mean = 4.7 out of 5; SD = 0.48).When asked about its applicability to real clinical settings, all participants (100%) agreed that the simulator could be effectively implemented as a training model in maxillofacial surgery residency programs, with the highest possible score in this item (mean = 4.8 out of 5; SD = 0.42).A total of 70% of respondents strongly agreed that the simulator achieved an appropriate level of realism for practicing fracture reduction, reconstruction, and fixation in this anatomical region (mean of 4.2 out of 5; SD = 0.92).Furthermore, 80% of participants agreed or strongly agreed that the simulator was suitable for learning the application of specific fixation systems, such as screws, plates, and meshes, relevant to the treatment of orbitozygomatic fractures (mean = 4.3 out of 5; SD = 1.34).Regarding its educational utility, 80% of participants strongly agreed that the simulator represents a valuable tool for training residents in performing reconstruction, reduction, and fixation procedures for orbitozygomatic fractures (mean = 4.7 out of 5; SD = 0.48).Lastly, 70% agreed or strongly agreed that the model is also useful for training surgeons with prior experience in maxillofacial trauma, highlighting its potential for continuing surgical education (mean = 3.7 out of 5; SD = 1.34).

#### 3.4.2. Face Validity

Internal consistency for the face validity questionnaire was assessed using Cronbach’s alpha, yielding a value of 0.790, which is considered acceptable reliability for the instrument.

The satisfaction survey results are shown in Table 11.

The results of the satisfaction survey reflected a high perceived level of realism among participants regarding the evaluated simulator.

A total of 90% of respondents agreed or strongly agreed that the visual appearance of the blow-out fracture was realistic, with a mean score of 4.3 out of 5 (SD = 0.95).Similarly, 80% of participants agreed or strongly agreed that the visual representation of the frontozygomatic fracture appeared realistic, with a mean score of 4.3 out of 5 (SD = 0.82).Regarding anatomical representation, 100% of participants agreed or strongly agreed that the phantom accurately simulated the depth and anatomy of the orbitozygomatic region, achieving a mean score of 4.6 out of 5 (SD = 0.52).For soft tissue realism, 70% of participants reported that soft tissues felt realistic during the minimally invasive surgical approach, with a mean score of 3.7 out of 5 (SD = 0.82).In terms of bone structure simulation, 60% of respondents strongly agreed that both cortical and cancellous bone provided a realistic tactile experience during drilling and screw placement, with a mean score of 4.1 out of 5 (SD = 1.29).

Finally, overall acceptance of the simulator: 100% of participants stated that they would use the model as a practice tool prior to performing real orbitozygomatic fracture surgery. This statement also received the highest rating, with a mean score of 4.6 out of 5 (SD = 0.52), reinforcing the simulator’s perceived educational value and clinical relevance.

### 3.5. Cone-Beam Tomography Ancillary Assessment (CBCT)

From this evaluation, four phantoms achieved a flawless outcome, with no errors observed in any of the assessed characteristics; all belonged to the expert group. In contrast, within the novice group, two cases were noteworthy. One case failed to achieve optimal coverage of the bony defect, resulting in only partial coverage. This incomplete coverage led to localized soft tissue infiltration beneath the mesh–bone interface, which subsequently allowed orbital fat herniation into the maxillary sinus cavity.

Additionally, a second novice case showed superior displacement of the reconstruction mesh, secondary to inadequate adaptation along the lateral orbital wall. This mispositioning resulted in soft tissue interposition at the mesh–bone interface and subsequent partial orbital fat protrusion into the maxillary sinus cavity. In both cases, the novices’ performance was classified as unfavorable surgical outcomes.

The dataset supporting the findings of this study has been published and is publicly available in the Dataverse repository.

## 4. Discussion

As shown in Section 3.5 of the Results, novice participants may commit surgical errors during their learning process which, if occurring in real patients, could lead to unfavorable outcomes or postoperative complications—potentially even requiring revision surgery to correct defects. This underscores the critical importance of initial training on phantoms before transitioning acquired skills to real surgical environments.

Clinical skills are typically acquired more rapidly than manual surgical skills, which require extensive training and practice. Consequently, maxillofacial trainees must develop a combination of theoretical knowledge and hands-on surgical experience [30].

Simulation-based education has been shown to effectively transfer technical skills to the operating room [31]. Additionally, previous studies have reported that simulation programs reduce learning curves [4,9,32], lower surgical training costs [33], and minimize complications associated with medical errors [20,34,35]. Given these advantages, the development of this simulation model may facilitate the practice of key steps in orbitozygomatic fracture reconstruction, particularly for novice trainees, shortening the learning curve.

In this study, we developed a synthetic and anatomically accurate orbitozygomatic fracture reconstruction simulation model as a training tool for maxillofacial surgery trainees. This approach enables the production of multiple identical phantom models, each with the same fractures and preoperative conditions, ensuring repeatability and consistency. These characteristics make the model particularly suitable for maxillofacial training programs, serving both as a training tool and an objective assessment platform for maxillofacial surgery skills.

To the best of our knowledge and based on previously published findings included in patent No. 3766-2024 [36], published on February 14 in the Chilean Official Gazette (CVE 2609714), which is provided as an annex to this manuscript, no other orbitozygomatic fracture simulation model combining both soft and hard tissues currently exists.

This high-fidelity model may be particularly beneficial for junior residents, allowing them to acquire basic competencies more rapidly and take greater advantage of real surgical opportunities [37]. The results of this study demonstrate that training with this simulation model exhibits construct validity, as it successfully differentiates between varying levels of surgical expertise. Specifically, SFM scores and operative time revealed statistically significant differences between faculty members and trainees, underscoring the model’s potential as an objective assessment tool for orbitozygomatic fracture reconstruction skills.

All study participants agreed that the phantom accurately simulated the anatomy of the orbitozygomatic region and reported they would use the simulator for practice prior to performing the procedure in real-life surgical scenarios. The mean scores exceeded 4.6 out of 5, indicating high levels of user satisfaction and supporting the model’s face validity.

Regarding content validity, all participants evaluated the model’s educational effectiveness, stating they would recommend it to peers and support its implementation in maxillofacial surgery residency programs. Additionally, all participants rated the model as a valuable tool for training residents in performing reconstruction procedures for orbitozygomatic fractures, with mean scores above 4.7 out of 5, further demonstrating strong user satisfaction and content validity.

Our findings regarding the model’s construct validity, evidenced by significant differences in OSATS scores, SFM (error count), and operative time between faculty and trainees, are consistent with validated simulation training systems across various surgical specialties. For instance, a high-fidelity model for minimally invasive hallux valgus surgery effectively differentiated experts from novices based on Objective Structured Assessment of Technical Skills (OSATS) scores and surgical time [38]. Similarly, a simulation module for ileo-transverse intracorporeal anastomosis demonstrated construct validity through significant differences in OSATS scores and operative times between experience levels [39]. In laryngeal microsurgery, a synthetic model successfully distinguished experts from residents using global rating scales (GRSs), specific rating scales (SRSs), execution time, and path length [22]. Furthermore, a model for endoscopic rectus sheath plication showed that experts achieved significantly higher OSATS and procedure-specific checklist scores than non-experts [40]. These consistent findings across diverse surgical fields underscore the reliability and utility of advanced simulation models in objectively assessing technical skills and differentiating between levels of surgical expertise.

Thus, such validated models actively contribute to improving training standards across surgical specialties, with scientific validation serving as a foundational step in the development of innovative educational strategies within residency programs, both for teaching and for the summative assessment of trainees.

Beyond construct validity, these studies are increasingly complemented by evaluations of user satisfaction and perceived realism, as these features are essential for engaging both trainees and instructors while enhancing the authenticity of surgical simulation. For example, the laryngeal microsurgery simulation system and the endoscopic rectus sheath plication model both achieved high scores for face and content validity among participating experts [22,40]. Similarly, evaluation of the hallux valgus MIS model revealed high satisfaction with its anatomical representation, handling characteristics, and utility as a training tool [38]. Collectively, these findings reinforce the critical role of simulation in modern surgical education, particularly for complex procedures such as orbitozygomatic fracture repair.

In an era where surgical exposure for residents must be regarded as a limited and valuable resource, simulation provides an accessible and ethical alternative for skill acquisition [41] prior to real surgical procedures. It enables trainees to practice fundamental skills in a controlled environment, ensuring that their initial operative experiences focus on advanced tasks, having already mastered basic techniques on the simulator. This ethical and cost-effective approach to surgical skill acquisition is rapidly evolving across all areas of medicine, and maxillofacial surgery should be no exception. It is expected to become increasingly widespread, supported by multiple studies demonstrating its effectiveness even in unsupervised settings with remote asynchronous feedback [42].

The integration of detailed feedback mechanisms, such as our Specific Fault Measurement (SFM) scale, further enhances its educational value by providing targeted areas for improvement and fostering personalized learning experiences [43]. Looking ahead, this approach aligns seamlessly with broader advancements in educational technologies, including the proven effectiveness of unsupervised simulation with remote asynchronous feedback [42] and the emerging potential of artificial intelligence to provide valuable, objective feedback in surgical scenarios [44]. This evolution in training allows residents to develop proficiency more efficiently, potentially reducing learning curves [38] and boosting self-confidence [41] in a safe, repeatable environment.

### 4.1. Specific Rating Scale Validation

To ensure a robust evaluation framework, we also developed a Specific Rating Scale (Specific Fault Measurement (SFM) Scale) based on OSATS-defined criteria.

A strong negative correlation was observed between SFM and OSATS scores. Given the significance level (*p* < 0.05), the null hypothesis was rejected, indicating that a high-intensity inverse relationship exists between the two scores. This result supports the convergent validity of the newly developed scale, which aligns with the conceptual framework of both assessment tools. While OSATS scores reflect overall surgical proficiency—with higher scores indicating better performance—the SFM scale quantifies technical errors, where higher scores indicate a greater number of procedural faults; conversely, participants with higher OSATS scores—indicative of superior surgical performance—committed fewer errors on the SFM scale.

Given that the SFM provides granular and domain-specific feedback, it serves as a valuable tool for post-simulation debriefing, helping educators identify precise areas for improvement and tailor feedback more effectively.

Concurrent validity was demonstrated by a strong inverse correlation between SFM and OSATS scores, and construct validity was supported by the significant differences observed between the faculty group and trainee group. Additionally, internal consistency, which measures the reliability of different items within a composite score, was high for both the SFM and OSATS, indicating that all assessment items contribute meaningfully to the final score.

The traditional apprenticeship model remains the standard method for training surgeons in orbitozygomatic fracture reconstruction, wherein inexperienced surgeons operate on live patients under supervision [24,45].

However, this traditional approach presents significant ethical, logistical, and financial challenges, making it increasingly difficult to justify as the sole training method in modern surgical education.

Moreover, previously developed simulation training models for the maxillofacial region are often low-fidelity, which results in limited educational outcomes and very limited effectiveness in replicating real surgical conditions [8,14,15,16,17,18,19,20,46,47]. Alternatively, high-fidelity simulation methods, such as the use of cadaveric specimens, while offering anatomical realism, suffer from major cost-related limitations, with some reports indicating costs exceeding USS 1500 per simulation station used by just two trainees [4].

This synthetic model offers a cost-effective, reproducible, and ethically viable alternative to traditional training approaches, avoiding the drawbacks associated with cadaveric, animal, or virtual reality models.

Despite its advantages, some limitations of the model were noted. One concern raised by participants was the lack of intraoperative bleeding, which facilitated visualization of deep structures and potentially simplified the procedure. Future iterations of the model should incorporate mechanisms to enhance realism by simulating surgical bleeding.

Technical difficulties also affected data collection, as two of ten participants could not be evaluated using the ICSAD tracking system due to unforeseen malfunctions. Despite these exclusions, differences in hand movements and path length were still observed between the trainee and faculty groups.

Thus, a potential limitation lies in the influence of the assessment tools on surgical performance. The use of ICSAD motion sensors and first-person recording cameras, which are not typically employed in live surgical settings, may have affected participant performance. To mitigate this risk, all recording devices were carefully positioned before the procedure, ensuring that the surgeon was comfortable and that no loose or obstructive elements interfered with the surgical process, and pairs of one faculty and one trainee participant always used the exact same equipment to avoid introducing performance bias between groups. At the end, all participants reported feeling comfortable with the setup during surgery, except for one case, in which a motion sensor detached mid-procedure, resulting in the exclusion of that data from analysis. A second exclusion occurred due to a voltage drop that rendered the ICSAD system inoperable for the remainder of the simulation.

Additionally, this study acknowledges that the sample size was relatively small due to the limited availability of surgeons from participating institutions. This limitation indicates that the reliability of the performance-related conclusions is constrained and should be interpreted with caution. This point is further reinforced by our recommendation that future studies include larger sample sizes to enhance the generalizability of the findings. To minimize variability, a standardized assessment protocol was implemented, ensuring that all participants were evaluated under identical conditions using identical fracture models with consistent anatomical characteristics. This approach provided a level of uniformity that is impossible to achieve in live patient surgeries, where anatomical and fracture variations introduce significant confounding factors.

Similarly, a clear advantage arises over cadaveric simulation models—whether human or animal—as, although these can be prepared using standardized protocols, their inherent anatomical variability undermines sample uniformity. As noted in a previous study in otolaryngology simulation, notable differences in porcine ear specimens were observed, illustrating the challenges posed by biological variability in surgical training [48].

A notable strength of the present study is the consistency and predictability of the synthetic model, which enables repeated training and assessment under standardized conditions. As a result, the only differentiating variable in this study was the surgeons’ performance, thereby enhancing the reliability of the assessment outcomes.

### 4.2. Future Directions

Future studies should incorporate larger sample sizes to improve the generalizability of the findings. Additionally, further research is needed to explore learning curves, as well as the model’s external and predictive validity, through applicability studies that assess how skills acquired in simulation translate to real operating room performance, ultimately enabling generalization to patient-based outcomes.

This study is intended to lay the groundwork for a subsequent longitudinal investigation aimed at evaluating residents through repeated performance assessments across multiple sessions. This approach will allow us to accurately determine their learning curve and later assess the impact on their performance in a real clinical setting.

## 5. Conclusions

The orbitozygomatic fracture simulation model and SFM developed in this study serve as surgical assessment tools for maxillofacial training programs. This low-cost, synthetic model is widely accessible and can be integrated into maxillofacial surgical training programs for skill development and assessment.

In an era of increasing awareness of medical errors and patient safety, there is a pressing need for continuous training of both students and specialists. Ethical constraints prohibit trainees from learning complex procedures directly on patients, while cadaveric and animal-based simulators present inherent accessibility, ethical, and financial challenges. This synthetic model offers a feasible, reproducible, and ethical solution, addressing the limitations of traditional training approaches while providing an objective, standardized, and cost-effective platform for maxillofacial surgical education.

## 6. Patents

This simulation model was the winner of the LXX Concurso Patentar para Transferir at Pontificia Universidad Católica de Chile and has been granted patent No. 3766-2024 [36], published on February 14 in the Chilean Official Gazette (CVE 2609714).

## Figures and Tables

**Figure 1 cmtr-18-00034-f001:**
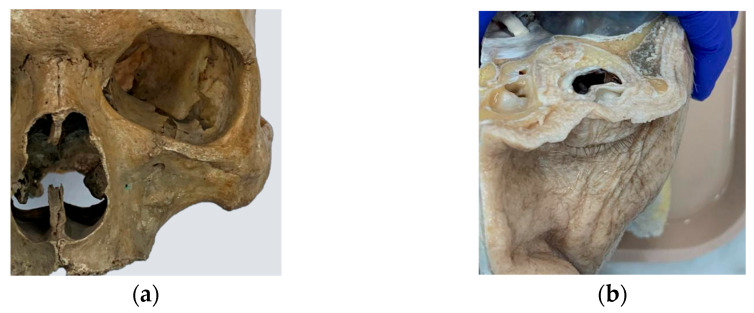
The design of the simulator was based on the analysis and observation of anatomical models from the anatomy laboratory of the School of Medicine at the Pontificia Universidad Católica de Chile, as well as in vivo procedures: (**a**) dry human skull specimen; (**b**) anatomical dissection of orbital soft tissues.

**Figure 2 cmtr-18-00034-f002:**
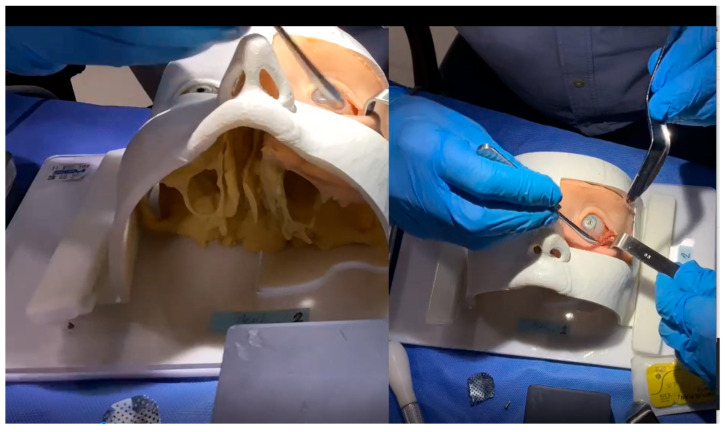
Screenshot from an actual training session video analyzed by experts. The video presents two simultaneous views of the phantom during the simulated surgery, allowing experts to evaluate the procedure without any visual obstruction at any point.

**Figure 3 cmtr-18-00034-f003:**
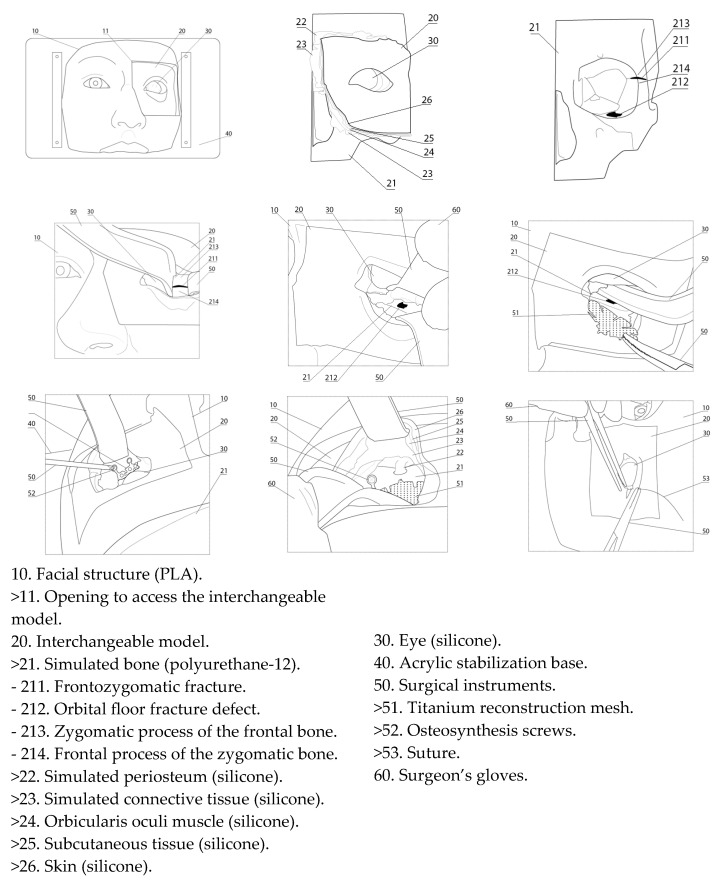
Description of the simulator components.

**Figure 4 cmtr-18-00034-f004:**
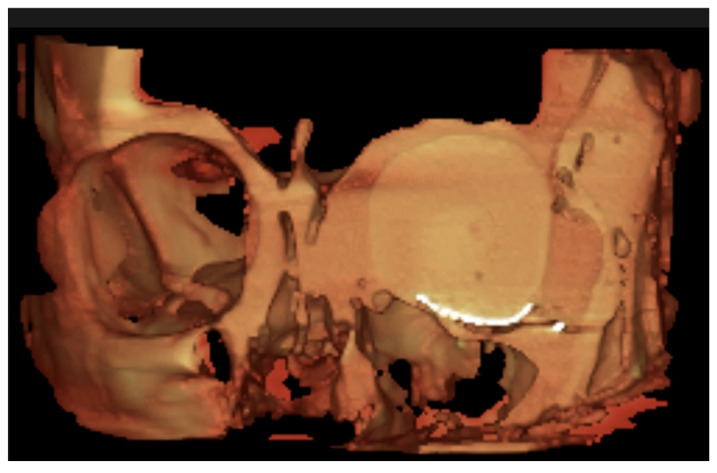
CBCT virtual rendering of the mesh, anterior coronal section.

**Figure 5 cmtr-18-00034-f005:**
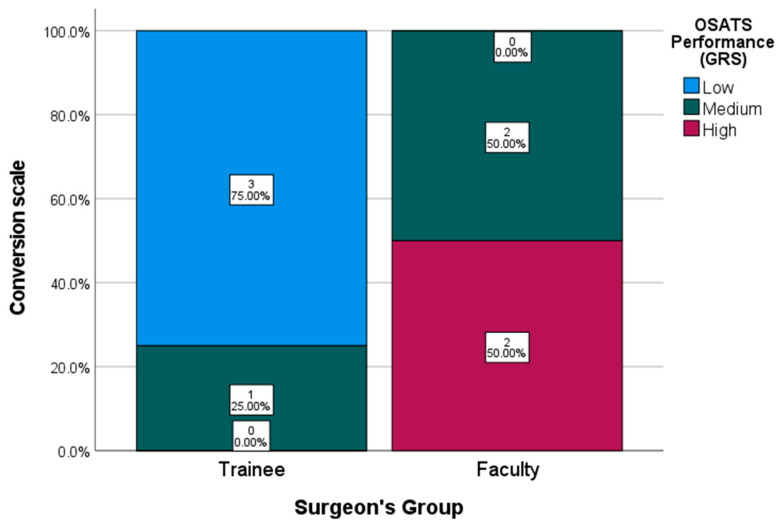
OSATS Performance Conversion Scale (GRS).

**Figure 6 cmtr-18-00034-f006:**
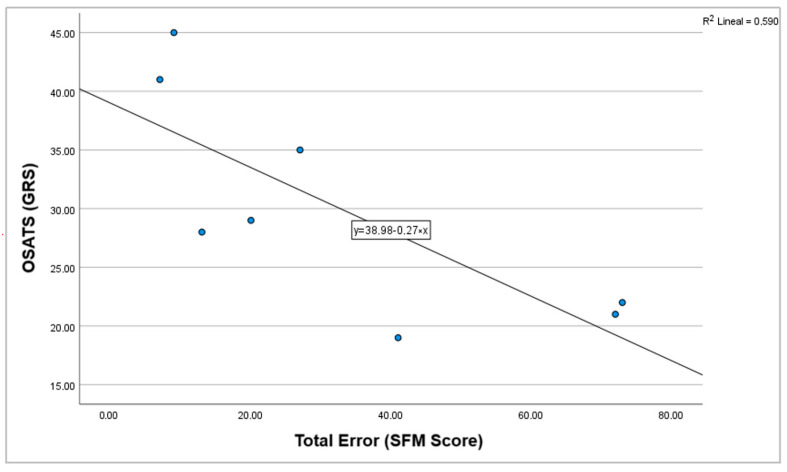
Correlation graph between OSATS (GRS) and total errors (SFM score).

**Figure 7 cmtr-18-00034-f007:**
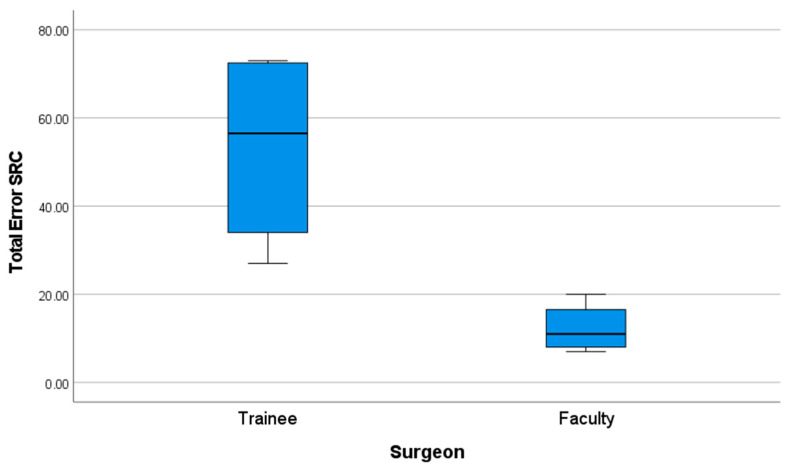
Performance graph of total errors (SFM score).

**Figure 8 cmtr-18-00034-f008:**
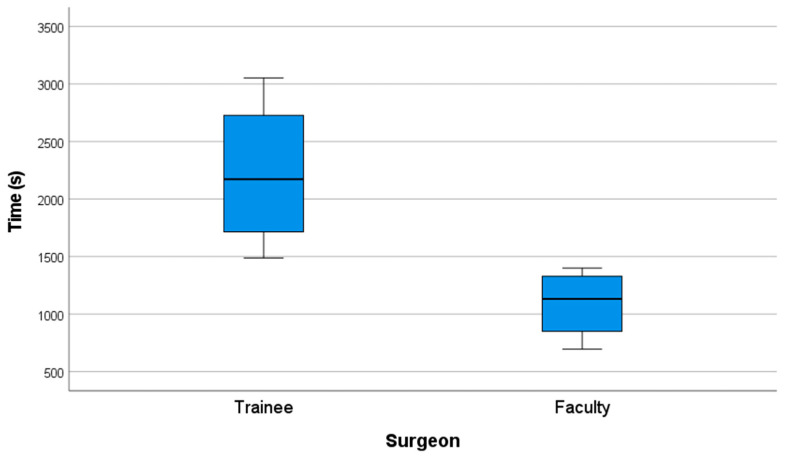
Performance graph of time (s).

**Figure 9 cmtr-18-00034-f009:**
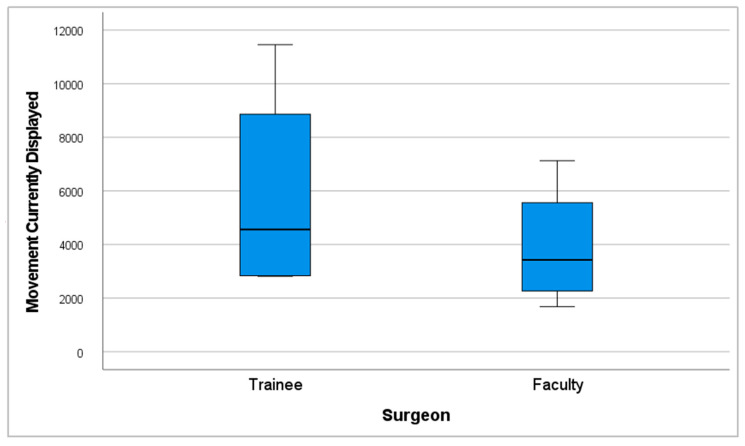
Performance graph of movements currently displayed.

**Figure 10 cmtr-18-00034-f010:**
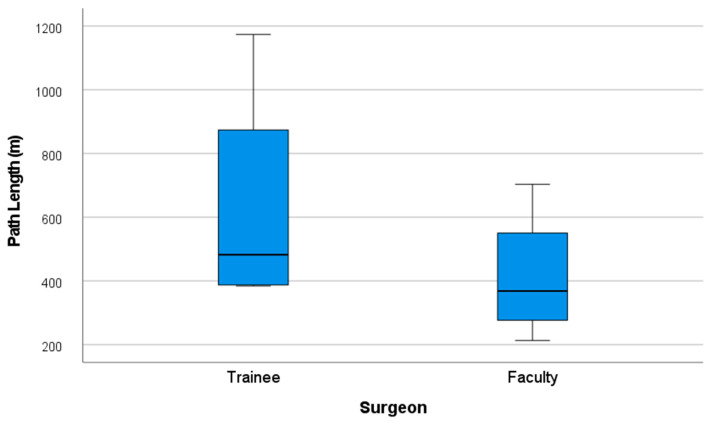
Performance graph of path length (m).

**Table 1 cmtr-18-00034-t001:** Preliminary questionnaire.

Initial Survey—Assessment of Surgical Experience 1. Level of Training in Oral and Maxillofacial Surgery and Traumatology: ☐ Resident ☐ Specialist 2. Years of Experience: How many years have you been practicing as a Resident or Specialist? ______ years. 3. Current Areas of Professional Practice: Mark with an X all the areas in which you are currently active: ☐ Teaching activities ☐ Non-surgical clinical practice ☐ Operating room activities ☐ Administrative duties 4. Career Interruptions: Please indicate whether you have had to suspend your clinical or surgical activities for a period longer than 6 months during your career: ☐ I have never suspended activities for more than 6 months ☐ I have suspended clinical/surgical activities for more than 6 months Reason: _______________________________ 5. Experience with Surgical Simulation: Have you previously performed surgical simulation using high-fidelity mannequins (phantoms)? ☐ Yes ☐ No 6. Knowledge of the Surgical Procedures to Be Simulated: Are you familiar with the surgical procedures that will be simulated during the practical session? ☐ Yes, I am familiar with them and have practiced them before ☐ Yes, I am familiar with them, but I have not had the opportunity to practice them ☐ No, I am not familiar with them 7. Practical Experience in Surgical Procedures: If you had to estimate the number of surgeries in which you have performed these procedures on real patients during your career, what would that number be? Number of surgeries: ______

**Table 2 cmtr-18-00034-t002:** Face validity questionnaire.

Face Validity Statement	Strongly Agree	Agree	Neutral	Disagree	Strongly Disagree
1. The visual appearance of the blow-out fracture looks realistic.					
2. The visual appearance of the frontozygomatic fracture looks realistic.					
3. The phantom accurately simulates the depth and shape of the orbitozygomatic anatomy.					
4. The soft tissues of the orbitozygomatic complex feel realistic when performing the minimally invasive surgical approach.					
5. The cortical bone and cancellous bone feel realistic when drilling and inserting screws.					
6. I believe I would practice with this simulator before performing an actual orbitozygomatic fracture surgery.					

**Table 3 cmtr-18-00034-t003:** Content validity questionnaire.

Content Validity Statement	Strongly Agree	Agree	Neutral	Disagree	Strongly Disagree
7. The simulator achieves overall realism for practicing reconstruction, reduction, and fixation of orbitozygomatic complex fractures.					
8. This simulator is a useful tool for residents to learn how to perform reconstruction, reduction, and fixation of orbitozygomatic complex fractures.					
9. This simulator is a useful tool for training surgeons with prior experience in maxillofacial trauma.					
10. You would consider implementing this simulator as a training model for maxillofacial surgery residents.					
11. This simulator can be used to learn how to perform surgical access to orbitozygomatic complex fractures.					
12. This simulator is suitable for learning how to use the screw, plate, and mesh systems specific to orbitozygomatic complex fractures.					

**Table 4 cmtr-18-00034-t004:** CBCT evaluated categories and their corresponding criteria.

Category	Evaluated Aspect
1. Mesh–bone interface.	Presence or absence of soft tissue at the mesh–bone interface.
2. Mesh/function.	Mesh position preventing or allowing orbital fat protrusion into the maxillary sinus.
3. Mesh/position.	Complete coverage of the orbital floor
4. Mesh adaptation.	Proper anatomical adaptation to the orbital cavity.
5. Mesh positional integrity.	Evidence of displacement, deformation, or rotation.
6. Relationship with maxillary sinus.	Presence or absence of sinus invasion.

**Table 5 cmtr-18-00034-t005:** Three-level performance conversion scale.

Performance Level	Score Range
Low	10–23
Intermediate	24–37
High	38–50

**Table 6 cmtr-18-00034-t006:** Surgical error record, organized by five assessment domains contributing to the SFM score.

Category	Type of Error Committed	Number of Errors Observed
1. Respect for Tissue	Violation of appropriate tissue handling.	
2. Unnecessary Movements	Performs unnecessary or non-purposeful movements.	
3. Hesitant or Clumsy Movements	Exhibits hesitation or clumsiness during any part of the procedure.	
4. Operative Flow	Interrupts or disrupts the surgical workflow.	
5. Procedural Knowledge	Demonstrates lack of procedural knowledge by making an error, expressing uncertainty, or requiring clarification on the next steps.	
**Total Errors (SFM Score)**	**Sum of all errors committed across categories.**	

**Table 7 cmtr-18-00034-t007:** Summary of participant demographic characteristics and clinical experience.

Group	n	Age, Mean (Range)	Years in Practice, Mean (Range)	No. of Orbitozygomatic Surgeries, Mean (Range)	No. of Surgeries per Year, Mean (Range)
**Trainees**	4	33 (29–36)	2.5 (2–4)	0.75 (0–3)	0.19 (0–0.75)
**Faculty**	6	47 (39–63)	14.8 (7–26)	351.67 (30–1600)	17.12 (4–61.53)

**Table 8 cmtr-18-00034-t008:** Specific Fault Measurement (SFM) test statistics.

Statistics				
Specific Fault Measurement (SFM)				
Trainees	N	Valid		4
		Missing		0
	Mean			53.25
	Std. Deviation			22.955
	Minimum			27
	Maximum			73
	Percentiles	25		30.50
		50		56.50
		75		72.75
Faculty	N	Valid		4
		Missing		0
	Mean			12.25
	Std. Deviation			5.737
	Minimum			7
	Maximum			20
	Percentiles	25		7.50
		50		11.00
		75		18.25
*Test Statistics ^a^*				
			Specific Fault Measurement (SFM)	
Mann–Whitney U			0.000	
Wilcoxon W			10.000	
Z			−2.309	
Asymp. Sig. (2-tailed)			0.021	
Exact Sig. [2(1-tailed Sig.)]			0.029 ^b^	

^a^. Grouping variable: surgeon. ^b^. Not corrected for ties.

**Table 9 cmtr-18-00034-t009:** Operative time comparison test statistics.

*Statistics*			
Time			
Trainees	N	Valid	4
		Missing	0
	Mean		2221.25
	Std. Deviation		668.360
	Minimum		1488
	Maximum		3053
	Percentiles	25	1601.50
		50	2172.00
		75	2890.25
Faculty	N	Valid	4
		Missing	0
	Mean		1089.75
	Std. Deviation		309.518
	Minimum		695
	Maximum		1400
	Percentiles	25	772.75
		50	1132.00
		75	1364.50
*Test Statistics ^a^*			
		Time	
Mann–Whitney U		0.000	
Wilcoxon W		10.000	
Z		−2.309	
Asymp. Sig. (2-tailed)		0.021	
Exact Sig. [2(1-tailed Sig.)]		0.029 ^b^	

^a^. Grouping variable: surgeon. ^b^. Not corrected for ties.

**Table 10 cmtr-18-00034-t010:** Table shows content validity results.

Content Validity Questions Statement/Answers (*N* = 10)	Strongly Agree	Agree	Neutral	Disagree	Strongly Disagree
This simulator can be used to learn how to perform surgical access to fractures of the orbitozygomatic complex.	7 (70%)	3 (30%)	0 (0%)	0 (0%)	0 (0%)
I would consider it appropriate to implement this simulator as a training model for maxillofacial surgery residents.	8 (80%)	2 (20%)	0 (0%)	0 (0%)	0 (0%)
The simulator achieves general realism for practicing reconstruction, reduction, and fixation of fractures of the orbitozygomatic complex.	5 (50%)	2 (20%)	3 (30%)	0 (0%)	0 (0%)
This simulator is suitable for learning how to use the screw, plate, and mesh system specific for fractures of the orbitozygomatic complex.	7 (70%)	1 (10%)	1 (10%)	0 (0%)	1 (10%)
This simulator is a useful tool for residents to learn how to perform reconstruction, reduction, and fixation of orbitozygomatic complex fractures.	8 (80%)	2 (20%)	0 (0%)	0 (0%)	0 (0%)
This simulator is a useful tool for training surgeons with previous experience in maxillofacial trauma.	3 (30%)	4 (40%)	1 (10%)	1 (10%)	1 (10%)

**Table 11 cmtr-18-00034-t011:** Table shows face validity results.

Face Validity Questions Statement/Answers (*N* = 10)	Strongly Agree	Agree	Neutral	Disagree	Strongly Disagree
The visual appearance of the Blow out Fracture looks realistic	5 (50%)	4 (40%)	0 (0%)	1 (10%)	0 (0%)
The visual appearance of the frontozygomatic fracture looks realistic.	5 (50%)	3 (30%)	2 (20%)	0 (0%)	0 (0%)
The model adequately simulates the depth and shape of the orbitozygomatic anatomy	6 (60%)	4 (40%)	0 (0%)	0 (0%)	0 (0%)
The soft tissues of the Zygomatic-orbital complex feel realistic when performing minimally invasive surgical access.	1 (10%)	6 (60%)	2 (20%)	1 (10%)	0 (0%)
Realistic cortical bone and cancellous bone feel when drilling and inserting screws	6 (60%)	1 (10%)	1 (10%)	2 (20%)	0 (0%)
Do you think you would practice with this simulator before an actual orbitozygomatic complex fracture surgery?	6 (60%)	4 (40%)	0 (0%)	0 (0%)	0 (0%)

## Data Availability

The dataset supporting the findings of this study has been published and is publicly available in the Dataverse repository. It can be accessed through the following: https://doi.org/10.60525/04teye511/CUDFRE. The dataset is licensed under the Creative Commons Attribution-ShareAlike 4.0 International License (https://creativecommons.org/licenses/by-sa/4.0/deed.es) (accessed on 3 June 2025), which permits sharing and adaptation with appropriate credit and under the same license terms. This open access availability supports transparency, reproducibility, and collaborative use within the research community.

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
