# Peer review of "Advanced Simulation System for Orbitozygomatic Fracture Reconstruction: Multicenter Validation of a Novel Training and Objective Assessment Platform"

_1943-3883, 2025, doi:10.3390/cmtr18030034_

Round 1

Reviewer 1 Report

Comments and Suggestions for Authors

  1. The manuscript's quality and data presentation are acceptable and important for clinicians and even patients.
  2. The manuscript advances our understanding of maxillofacial trauma and it managements.
  3. The title should be rewritten to be more precise and explanatory.
  4. Make the abstract more informative and it should represent the article's substance and be no more than 250 words long.
  5. Include four to six keywords that are relevant to the manuscript but not stated in the title.
  6. Additional paragraphs to introduce further studies onfacial trauma and its management .
    Suggested references:
  • Alshalah ME, Enezei HH, Aldaghir OM, Khalil AA, Aldelaimi TN, Al-Ani RM. Direct or Indirect Surgical Approach of Zygomatic Complex Fracture: A Comparative Study. J Craniofac Surg. 2023;34(8):2433-2436. doi:10.1097/SCS.0000000000009712
  • Aldelaimi TN. New maneuver for fixation of pediatric nasal bone fracture. J Craniofac Surg. 2011;22(4):1476-1478. doi:10.1097/SCS.0b013e31821d1997
  • Aldelaimi TN, Khalil AA. Surgical management of pediatric mandibular trauma. J Craniofac Surg. 2013;24(3):785-787. doi:10.1097/SCS.0b013e31828b6c47
  • Enezei, H.H., Khalil, A.A., Naif, T.N.. A Clinical Analysis of Surgically Managed Mandibular Fractures: Epidemiology, Clinical Profile, Patterns, Treatments, and Outcome. International Medical Journal 2020; 27(4): 1 - 4.

  1. Authors should check for writing and typing errors.
  2. The statements in discussion are acceptable but few paragraphs about the justification of your findings and comparison with other recent relevant studies.
  3. Only include up-to-date references in the reference list and remove outdated ones.

Good Luck

Author Response

Dear Reviewer 1,
We sincerely appreciate the time and effort you dedicated to reviewing our manuscript. Please find our detailed responses to your comments below.

1.    The manuscript's quality and data presentation are acceptable and important for clinicians and even patients.
2.    The manuscript advances our understanding of maxillofacial trauma and it managements.
3.    The title should be rewritten to be more precise and explanatory.

We reconsidered the title of our article and proposed three alternatives:

Advanced Simulation System for Orbitozygomatic Fracture Reconstruction: Multicenter Validation of a Novel Training and Objective Assessment Platform

Synthetic Simulation Model for Orbitozygomatic Fracture Repair: Multicenter Validation and Implications for Surgical Training and Skill Assessment

Anatomically Realistic Synthetic Simulation for Orbitozygomatic Trauma Surgery: Multicenter Validation of a Novel Training Tool with Objective Metrics

We ultimately selected the first option, as we believe it best aligns with the recommendations provided.

4.    Make the abstract more informative and it should represent the article's substance and be no more than 250 words long.

We rewrote the abstract with a focus on making it informative and representative of the article’s substance, while ensuring it does not exceed 250 words.

Abstract:
Orbitozygomatic fractures represent a complex surgical challenge. Given the urgent need for validated educational tools that surpass traditional learning models, this multicenter study developed and validated a novel synthetic advanced simulation model for the reconstruction of these fractures. The model integrates platinum-cured silicones and 3D-printed bony structures with prefabricated fractures, accurately replicating the anatomy and tactile properties of soft and hard tissues, including simulated herniation of orbital contents. To our knowledge, it is the only available synthetic model combining both tissue types for this training.

Ten participants (faculty and residents) completed simulated procedures. Technical performance was assessed using a hand motion tracking system (ICSAD), the global OSATS scale, and a task-specific error measurement scale (SFM). Statistically significant differences (P = .021) were observed in operative time and error count between novices and experts, confirming the model’s construct validity. Faculty completed the surgery in significantly less time (mean 18.16 minutes vs. 37.01 minutes for residents) and made fewer errors (mean 12.25 vs. 53.25). Face and content validity were strongly supported by participant surveys, with 100% stating they would use the simulator to practice before real surgery. A strong inverse correlation (r = –0.786, P = .021) between OSATS and SFM scores demonstrated concurrent validity.

This model enables ethical, repeatable, and cost-effective training, supporting its implementation into surgical curricula to enhance competence and provide objective skill assessment in orbitozygomatic trauma surgery.

5.    Include four to six keywords that are relevant to the manuscript but not stated in the title.

We have selected the following keywords, which are relevant to the manuscript but not included in the title: Surgical Education, Technical Skills, Synthetic Surgical Model, Performance Feedback, Patient Safety, and OSATS, reflecting the study’s focus on simulation-based training, objective skill evaluation, and educational strategies in surgical practice.

6.    Additional paragraphs to introduce further studies on facial trauma and its management .
Suggested references:
Alshalah ME, Enezei HH, Aldaghir OM, Khalil AA, Aldelaimi TN, Al-Ani RM. Direct or Indirect Surgical Approach of Zygomatic Complex Fracture: A Comparative Study. J Craniofac Surg. 2023;34(8):2433-2436. doi:10.1097/SCS.0000000000009712 sdsdf

We acknowledge this valuable suggestion. The recommended paper has been cited in the following section of the manuscript: “Reconstruction of orbitozygomatic fractures demands not only surgical proficiency but also specialized training in minimally invasive techniques [5, 6], which are essential for restoring function while minimizing aesthetic sequelae in the facial region(cite).”

7.    Authors should check for writing and typing errors.

We thoroughly reviewed the manuscript, addressing minor errors, including 1 misspelling, 1 terminology error, 8 hyphenation errors, and 6 punctuation errors.

8.    The statements in discussion are acceptable but few paragraphs about the justification of your findings and comparison with other recent relevant studies.

9.    Only include up-to-date references in the reference list and remove outdated ones.

We have added several paragraphs justifying our findings and comparing them with other studies, and we have included only the most recent and relevant references.

The paragraphs were inserted in section "4. Discussion," after the paragraph ending with "further demonstrating strong user satisfaction and content validity" and before the section "4.1 Specific Rating Scale Validation."

The paragraphs below:

Our findings regarding the model's construct validity, evidenced by significant differences in OSATS scores, SFM (error count), and operative time between faculty and trainees, are consistent with validated simulation training systems across various surgical specialties. For instance, a high-fidelity model for minimally invasive hallux valgus surgery effectively differentiated experts from novices based on Objective Structured Assessment of Technical Skills (OSATS) scores and surgical time [38]. Similarly, a simulation module for ileo-transverse intracorporeal anastomosis demonstrated construct validity through significant differences in OSATS scores and operative times between experience levels [39]. In laryngeal microsurgery, a synthetic model successfully distinguished experts from residents using global rating scales (GRS), specific rating scales (SRS), execution time, and path length [22]. Furthermore, a model for endoscopic rectus sheath plication showed that experts achieved significantly higher OSATS and procedure-specific checklist scores than non-experts [40]. These consistent findings across diverse surgical fields underscore the reliability and utility of advanced simulation models in objectively assessing technical skills and differentiating between levels of surgical expertise.

Thus, such validated models actively contribute to improving training standards across surgical specialties, with scientific validation serving as a foundational step in the development of innovative educational strategies within residency programs, both for teaching and for the summative assessment of trainees.

Beyond construct validity, these studies are increasingly complemented by evaluations of user satisfaction and perceived realism, as these features are essential for engaging both trainees and instructors while enhancing the authenticity of surgical simulation. For example, the laryngeal microsurgery simulation system and the endoscopic rectus sheath plication model both achieved high scores for face and content validity among participating experts [22, 40]. Similarly, evaluation of the hallux valgus MIS model revealed high satisfaction with its anatomical representation, handling characteristics, and utility as a training tool [38]. Collectively, these findings reinforce the critical role of simulation in modern surgical education, particularly for complex procedures such as orbitozygomatic fracture repair.

In an era where surgical exposure for residents must be regarded as a limited and valuable resource, simulation provides an accessible and ethical alternative for skill acquisition [41] prior to real surgical procedures. It enables trainees to practice fundamental skills in a controlled environment, ensuring that their initial operative experiences focus on advanced tasks, having already mastered basic techniques on the simulator. This ethical and cost-effective approach to surgical skill acquisition is rapidly evolving across all areas of medicine, and maxillofacial surgery should be no exception. It is expected to become increasingly widespread, supported by multiple studies demonstrating its effectiveness even in unsupervised settings with remote asynchronous feedback [42].

The integration of detailed feedback mechanisms, such as our Specific Fault Measurement (SFM) scale, further enhances its educational value by providing targeted areas for improvement and fostering personalized learning experiences [43]. Looking ahead, this approach aligns seamlessly with broader advancements in educational technologies, including the proven effectiveness of unsupervised simulation with remote asynchronous feedback [42] and the emerging potential of artificial intelligence to provide valuable, objective feedback in surgical scenarios [44]. This evolution in training allows residents to develop proficiency more efficiently, potentially reducing learning curves [38] and boosting self-confidence [41] in a safe, repeatable environment.

Finally, to facilitate your review, we have attached a copy of the revised text in which the modifications have been highlighted in color, according to the reviewers’ suggestions.

Kind regards,
The authors

Reviewer 2 Report

Comments and Suggestions for Authors

The article is written in-depth and requires appreciation.

Author Response

Dear Reviewer 2,

We sincerely thank you for your review.

Kind regards,
The authors

Reviewer 3 Report

Comments and Suggestions for Authors

Dear Authors,

Thank you for the opportunity to review your manuscript. The topic you address is highly relevant and contributes meaningfully to the evolving field of surgical education. High-fidelity simulation models have become increasingly important as surgical training transitions toward ethical, standardized, and competency-based frameworks. Your proposed simulator offers a commendable attempt to advance maxillofacial training and could serve as a foundation for more refined educational platforms in the future.

That said, while the concept and intention of the study are strong, the manuscript requires further development in several important areas before it can be considered for publication:

  1. Lack of Objective Accuracy Assessment
    Despite the use of observation-based evaluation methods, the study lacks an objective assessment of the anatomical accuracy of fracture reduction or implant positioning. Quantitative outcome measures such as CT comparison or 3D model superimposition would significantly enhance the scientific validity of the simulator and demonstrate its value beyond basic technical training.
  2. Small Sample Size and Missing Data
    With only 10 participants and incomplete motion tracking data in two cases, the statistical power of the findings is limited. As a result, the reliability of the performance-related conclusions is somewhat constrained and should be interpreted with caution.
  3. Absence of Predictive or Longitudinal Validation
    While the observation that faculty outperformed trainees is reassuring and suggests the system can differentiate levels of expertise, this outcome is expected. A more informative approach would be to assess trainees across multiple sessions using varied fracture scenarios to determine whether performance improves with repeated use - thereby evaluating the simulator's true educational impact.
  4. Simplified Soft Tissue Dynamics
    Although the model includes soft tissue layers, the manuscript does not adequately reflect the challenges involved in replicating realistic soft tissue properties. These aspects should be more clearly described in the Methods section."

Figure Comments – Figure 1:

  • Why does the specimen not reflect the orbital fracture, which is central to the study's aim?
  • The orbital anatomy in the image is unclear and could benefit from annotation or improved visualization.
  • The fixation plate appears to rest on soft tissue rather than directly on bone.

In conclusion, the study presents a valuable concept and addresses a clear need in the field. However, the above points should be addressed comprehensively in a revised version to support the manuscript’s publication and to more clearly demonstrate the accuracy, educational benefit, and translational potential of the simulation platform.

Author Response

Dear Reviewer 3,

We sincerely appreciate the time and effort you dedicated to reviewing our manuscript. Please find our detailed responses to your comments below.

1.    Lack of Objective Accuracy Assessment
Despite the use of observation-based evaluation methods, the study lacks an objective assessment of the anatomical accuracy of fracture reduction or implant positioning. Quantitative outcome measures such as CT comparison or 3D model superimposition would significantly enhance the scientific validity of the simulator and demonstrate its value beyond basic technical training.

Response: We incorporated an objective outcome assessment by comparing results using CBCT imaging and applying a structured radiological evaluation framework. This analysis allowed us to determine whether the surgical objectives were adequately achieved.

2.    Small Sample Size and Missing Data
With only 10 participants and incomplete motion tracking data in two cases, the statistical power of the findings is limited. As a result, the reliability of the performance-related conclusions is somewhat constrained and should be interpreted with caution.

Response: We acknowledge that the statistical power of our findings is limited. Accordingly, we have revised the statements in the Discussion section to ensure it is clearly conveyed that the reliability of the performance-related conclusions is restricted and should be interpreted with caution.

3.    Absence of Predictive or Longitudinal Validation
While the observation that faculty outperformed trainees is reassuring and suggests the system can differentiate levels of expertise, this outcome is expected. A more informative approach would be to assess trainees across multiple sessions using varied fracture scenarios to determine whether performance improves with repeated use - thereby evaluating the simulator's true educational impact.

Response: Indeed, this study is intended to lay the groundwork for a subsequent longitudinal investigation aimed at evaluating residents through repeated performance assessments across multiple sessions. This approach will allow us to accurately determine their learning curve and later assess the impact on their performance in a real clinical setting. In response to this point, we have revised the manuscript to ensure that this perspective is clearly articulated.

4.    Simplified Soft Tissue Dynamics
Although the model includes soft tissue layers, the manuscript does not adequately reflect the challenges involved in replicating realistic soft tissue properties. These aspects should be more clearly described in the Methods section."

Response: We have revised the Methods section to provide a more detailed description of the methodology used for replicating soft tissues. In this update, we elaborated on the materials selected, the layering techniques applied, and the considerations taken to approximate the biomechanical properties of soft tissues as closely as possible. We believe that this additional information will help readers better understand the challenges involved in soft tissue simulation and the technical decisions that guided our approach.
Figure Comments – Figure 1:
•    Why does the specimen not reflect the orbital fracture, which is central to the study's aim?
•    The orbital anatomy in the image is unclear and could benefit from annotation or improved visualization.
•    The fixation plate appears to rest on soft tissue rather than directly on bone.

In response to this comment, we decided to remove the figure from the revised manuscript, as it could potentially lead to reader confusion and to ensure a clearer and more coherent presentation.

Finally, to facilitate your review, we have attached a copy of the revised text in which the modifications have been highlighted in color, according to the reviewers’ suggestions.

Kind regards,
The authors

Round 2

Reviewer 1 Report

Comments and Suggestions for Authors

Dear authors

Thanks for taking in consideration the suggested comments and revisions.

Good Luck